# How Do Different Urban Footpath Environments Affect the Jogging Preferences of Residents of Different Genders? Empirical Research Based on Trajectory Data

**DOI:** 10.3390/ijerph192114372

**Published:** 2022-11-03

**Authors:** Qikang Zhong, Bo Li, Yue Chen

**Affiliations:** School of Architecture and Art, Central South University, Changsha 410083, China

**Keywords:** urban footpath, footpath environmental differentiation, gender, jogging activities, trajectory data

## Abstract

In recent years, the impact of the urban environment on residents’ physical activity (PA) has received extensive attention, but whether this impact has differences in the jogging preferences of residents in different footpath environments and different genders requires further research. Therefore, based on jogging trajectory data, this paper uses the grouping multiple linear regression model to study the different influencing factors of different footpath environments on the jogging of residents of different genders. The results show that (1) jogging activities (JA) were mainly concentrated in the community footpath environment, and its peak was reached at night; (2) the rise and fall of elements in built environments, social environments, and natural environments significantly affected the relative jogging distance of residents; (3) Residential land density (RLD) has a positive impact on the JA of community and green land footpaths and has a negative impact on the JA of urban footpaths. However, arterial road density (ARD) and bus distance density (BDD) have opposite significant effects on the JA of communities and green land footpaths; (4) ARD has the significant opposite effect on the JA for residents of different genders on urban footpaths and community footpaths. Facilities diversity (FD), population density (PD), and bus stop density (BSD) also had significant opposite effects on the JA of residents of different genders on green land footpaths. In general, we put forward a method theory to identify the footpath environment and provide references for improving the layout and construction of different gender residents for different footpath environment elements.

## 1. Introduction

At present, chronic diseases have become the number one factor threatening the health of Chinese residents [1]. Leisure-time simple physical activity (LTPA) is beneficial to health strengthening and chronic disease reduction, as well as heart and lungs [2], and a positive and friendly built environment can promote residents’ participation in physical activities (PA) to enhance their health [3]. Renalds et al. (2010) argue that the built environment is the foundation of health protection. By reviewing 23 relevant literature studies, they found that when the walkability of the community is higher, the frequency of the PA of residents will be higher, and the prevalence of obesity and depression will be lower [4]. The study of Frank et al. (2022) showed that community walkability and high-density park areas were significantly correlated with reducing the incidence of diabetes in residents [5]. In addition, Professor Xiao (2022) explored the health effects of the built environment on the elderly population and found that facility density and facility accessibility can affect the body mass index of the elderly [6]. Additionally, the built environment has been used as a research object for the spatial planning of residential areas by spatio-temporal geographers and urban planning scholars [7,8]. Therefore, the built environment has apparent effects on PA, which has become a hot issue concerned by relevant disciplines. Jogging has been proven to be one of the most beneficial daily exercise modes for the public [9,10,11]. Chakravarty et al. (2008) conducted a questionnaire survey on 538 members of American running clubs and 423 healthy people aged 50+ from Northern California (control group). Studies have shown that runners are younger, leaner, and less likely to smoke compared to controls. The death risk rate of runners was lower than that of the control group [9]. Wang et al. (2013) found that jogging was not only significantly negatively correlated with the risk of male death but also had a similar negative correlation with cancer and cardiovascular disease mortality by tracking 61,477 men in Shanghai for regular exercise [10]. Schnohr et al. (2015) interviewed 1098 healthy joggers and 3950 healthy non-joggers in Copenhagen. The results showed that the mortality risk of mild and moderate joggers was lower than that of sedentary non-joggers [11]. However, previous studies have focused on analyzing the advantages and disadvantages of the built environment for living, workplace, and activity destinations [12,13] as well as recreation, walking, and cycling [14,15]. Additionally, relatively little research has been conducted on jogging and footpath environments. The segregation of urban footpath environments formed by different mobility levels and spatial layouts is also seldom considered [16].

Meanwhile, studies usually focus on the different influences of the same built environmental factors on different groups [17,18,19,20,21], and fewer scholars have noticed footpath environment segregation. The main facilities and places in cities are distributed along the road network, and there are differences in the demand for facility service for different groups of people. The footpath environments vary according to facilities and places in different ways [22,23]. Urban roads are indispensable “tools” for residents to travel, providing opportunities for people to visit various activity sites. To some extent, it is a kind of mobile activity space. Therefore, footpath environment differentiation will also lead to differences in the jogging trajectory of residents, which will lead to different influencing elements of jogging [24]. Studying the differentiation of footpath environments is helpful to understand the differentiation of residents’ jogging activity (JA) and improving the activity footpath environments of residents, so as to promote residents’ participation in JA and provide a theoretical reference and basis for related practice.

Therefore, it is worth studying the effect of footpath environment differentiation on the JA of residents. In addition, the environmental differentiation of footpaths is influenced by social and natural factors. In addition, Ball et al. (2001) conducted a questionnaire survey of 3392 Australian adults and found that perceived environmental aesthetics and walking companions are important factors related to walking exercise in Australian adults [25]. Karusisi et al. (2012) found that high social cohesion can also increase the frequency of jogging [26]. In addition, Chen and Lin surveyed 5222 adults in urban areas of mainland China, Taiwan, and South Korea and found that in the Southeast Asian country, gender, marital status, education, and social status are all common factors for residents not to engage in PA. For the Chinese in particular, the perceived quality of urban communities is particularly important for maintaining physical activity. In addition, they also found that air quality is also a key factor affecting residents’ PA [27]. Liu et al. found that climate has a certain impact on jogging activities (JA). The number of joggers in urban parks in spring and summer is more than that in autumn and winter [28]. In addition, Han et al. also found that contact with the natural environment during jogging can improve the insight of athletes [29]. Jansen et al. also confirmed that the natural environment has a significant impact on JA [30]. Therefore, this study added the elements of social [31,32] and natural environments [30,33] on the basis of the built environment [34,35,36] to investigate in depth the differential effects of different footpath environments on residents’ JA. At the same time, this mechanism may have gender differences, thus affecting the different jogging trajectory distributions in the footpath network. Therefore, the different spatial constraints and individual needs of different gender groups in their jogging activities are important perspectives and theories to explain gendered urban space [37]. For example, Basu et al. conducted a series of online experiments to observe and evaluate environmental scenarios with 995 participants in Australia and found that residential, commercial, and mixed land provided pedestrians with a sense of security and had a greater impact on women [38]. Li et al. conducted a sample survey on 3360 elderly people aged 60+ in Beijing. The results found that shorter distances from bus stops can promote social interaction and health effects in elderly men (over 80 years old), and a high-quality built environment only significantly promotes PA in elderly women to reduce loneliness [39]. So far, questionnaire data and log surveys [40,41,42] are the major means to study gender differences with a lack of large sample analysis based on natural experiments. There is little discussion on gender segregation and its macro-effects produced by a large amount of daily JAs. For this, the spatio-temporal data analysis provides a new research method. Relevant studies have been conducted to reveal the social differentiation between the places of residence and places of activity and found that they have spatiotemporal effects at different scales and change to varying degrees with the activities of residents [43,44,45,46]. However, there is a lack of studies on the spatial differentiation of city footpath environments based on jogging, of which there are even fewer studies related to gender differentiation. Hence, it is necessary to reveal whether there are differences in the effects of different footpath environments on the JA in different gender groups.

In conclusion, this paper reveals the spatial differentiation of different gender groups’ JAs in the city footpath network and tries to answer the following main questions: ① Can the built environment, social environment, and natural environment around the jogging trajectory influence the JA of residents? ② Does the urban environment have any difference on JA for people in different synchronous footpath environments? ③ In reality, is there an objective gender differentiation phenomenon in the footpath network for individual jogging trajectory mapping? Is there any difference in the influence of the disparate footpath environment on the JA of different gender groups?

To address the above questions, this paper constructs a theoretical hypothesis analysis framework from the perspective of spatio-temporal, health geography, and daily activity space theory. Gender and space have been the focus of spatio-temporal geography for a long time. It is mainly believed that different genders suffer from rule constraints and behavioral restrictions from different spatial environments, including public and private spaces, and thus experience gender segregation in different scenarios, such as public facilities, accessibility, and commuting [47,48,49]. The phenomenon may be further extended to the city footpath network as a special public space, forming the gender-based social differentiation of the city footpath network. This differentiation may be influenced by the differences in JA between genders caused by different footpath environments, and then mainly manifested in the utilization of the space environment along the route footpaths. Therefore, we firstly defined the three footpaths: “urban footpath” means a footpath that only people can walk on in urban areas; “community footpath” includes roadways and sidewalks in community areas; “green land footpath” refers to all kinds of paths in green land park areas. On this basis, the theoretical hypothesis framework of this paper is constructed, and the following theoretical hypotheses are proposed (Figure 1): Firstly, the footpath environment of the movement trajectory is the main spatial carrier of residents’ JA, so the built environment, social environment, and natural environment around the jogging footpath have a positive or negative influence on residents’ JA (H1); Secondly, different footpath types of environments have different effects on JA (H2); Meanwhile, the distribution of trips in the footpath network of different gender groups shows relatively obvious differences. Different genders have different behavior decision-making that decides various activity patterns. Similarly, the social space differentiation phenomenon forms in the activity space and jogging footpath network. Therefore, different footpath environments have different effects on the JA of different gender groups (H3).

## 2. Data Sources and Research Methods

The central area of Shanghai is taken as a case study in this paper. Firstly, based on the multiple linear regression analysis method, the influence of the footpath environment of sports trajectory on the JA is explored; secondly, the group regression analysis method is adopted to reveal the influence of the environment around the different footpaths on the spatial differentiation of jogging trajectory; finally, the similarities and differences of the influence of the environment around the different footpaths on the jogging distribution of different gender groups are further compared and analyzed, so as to verify the above theoretical hypothesis.

### 2.1. Research Area and Research Data

In this paper, Shanghai is selected as the case study area. As one of the four mega cities in China, Shanghai has the highest level of socio-economic development in the country all year round, a large resident population, and a complex urban built environment with rapid development and changes. Therefore, the city is worthy of study and concern. The study area mainly includes Baoshan, Hongkou, Huangpu, Jiading, Jing’an, Minhang, Putuo, Songjiang, Xuhui, Yangpu, Changning, and Pudong New Area. The study data obtained from the Dorray sports app reflect the jogging activities of residents (Figure 2), and the website of this app is “ http://www.12sporting.com/ (accessed on 1 October 2020)”. In addition, the data in the study also include the use type of 2018 Shanghai land, road network, building vector, NDVI, natural environment, Point of Interest (POI) data, and a 30 m precision kilometer grid dataset of population spatial distribution (data from Geographic Remote Sensing Ecology Network: www.gisrs.cn (accessed on 3 June 2022).

This study used the Dorray sports app operated by smartphone to replace traditional methods (such as questionnaires, on-site interviews, and accelerometers) to obtain objective personal trajectory data. This new method is more convenient and can accurately record the time, distance, and location of the PA of residents anytime, anywhere, and is suitable for large-scale research and the long-term dynamic detection of PA, thus effectively making up for the shortcomings of traditional methods [50,51]. In order to improve the data accuracy, this research distinguished and filtered the data in this way: after obtaining the original data from the company of the Dorray Sport app, we obtain the data located in Shanghai through Python programming. The collected trajectory .txt file contains trajectory point coordinate information. Each .txt file contains (X, Y) geographic longitude and latitude coordinate information ranging from 10 to 1000 bits according to the trajectory length. The first step is to convert these coordinates into trajectory points in ArcGIS 10.1 (Environmental Systems Research Institute, Inc., RedLands, CA, USA) and then connect them into motion trajectories (Figure 3). The second step is to obtain effective motion trajectories after batch trajectory switching operations through the above process, and then link the attribute table in ArcGIS software according to the motion user ID of each trajectory to match the relevant motion information. The third step is to filter the trajectory data within the research scope through the “intersection” tool of the ArcGIS software. Invalid trajectories with abnormal motion factors such as position offset, short motion time (<1 min), short active distance (<0.1 km), and slow speed (<5 km/h) (the abnormality of speed is defined according to the minimum jogging speed criterion [52,53,54]) are eliminated. In addition, JA recognition: participants can record their sports information by selecting jogging mode on the app interface. In the remaining dataset, users with records of urban footpaths, community footpaths, and green land footpaths were identified and screened to be used as the study sample (all private data were removed to protect personal privacy). The processed dataset contained 8311 valid trajectory data with a sex ratio of 2.11 (the time period of the datasets: 4 September 2016–30 December 2018).

### 2.2. Indication System of Influencing Factors

According to existing studies, it has been pointed out that the built environment in different geographical backgrounds has an impact on the activity patterns and effects of different groups by constraining the corresponding travel activities [55]. For example, the travel activities of different genders are related to factors such as traffic accessibility, road design, and location conditions. Women travel for shorter distances and times and have a higher proportion of trips by public transportation or walking, especially in high-density, high-mix residential areas; men travel for longer distances and times, have a larger range of activities, with a high proportion of trips by cars [46,56,57,58]. The construction of this research environment system is based on the traditional 5D built environment framework [6,59]. Some studies have found that there is a certain correlation between the social and natural environment and residents’ activities and health [27,60,61,62]. Therefore, in this study, 17 indicators were selected as built environment factors for jogging footpaths: residential density, park greenfield density (reflecting the number of daily activity places for residents); main road density, secondary road density, and branch road density (reflecting the accessibility of residents to reach their activity places); facility POI density, population density, and nighttime light density (reflecting the degree of development and attractiveness of the area); site mix (reflecting the diversity of site functions and attractiveness of the area); precipitation density, temperature density, NDVI density, slope density (reflecting the comfort of the natural environment of the area); subway line length density, bus station density, distance from a bus station, and distance from a subway station (reflecting the convenience of transportation interchange for residents). The constructed environmental factors are measured as follows: according to the buffer radius criteria proposed by Frank [58], a buffer range based on the jogging trajectory with a radius of 500 m is created, and the density of each indicator is obtained by calculating the number or average value of each type of environmental indicator within the buffer area of each trajectory and dividing the average value by the buffer area. In addition, the total distance of the residents’ single JA was taken as the measurement indicator of PA in this study. The software starts data collection and records exercise data through the exerciser’s autonomous choice at the software interface entrance. Based on this, the indicator system of influencing factors was constructed (Table 1). Among them, the land use mixture was calculated in the following manner.
(1)H=−∑i=1NPilnPilnN

Note: H is the mixing indicator, $P_{i}$ is the ratio of the number i path type to the current buffer area, and N is the type of land use.

### 2.3. Analysis Method

In this paper, descriptive statistics are calculated with the help of SPSS 23, and then multiple linear regression models are used to explore the relationship between the variables and verify the proposed theoretical model at the same time. Firstly, the descriptive statistics and spatial visualization of residents’ JA characteristics are performed for showing the differences in the distribution of residents’ JA in different road networks for different gender groups. Secondly, regression analysis was applied to explore the relationship between the variables. Model 1 was constructed to explore the influence relationship between the footpath environment and the residents’ JA (test H1), model 2 to test the relationship between JA distribution and different footpath background environments (test H2), and model 3 to reveal the influence factors of JA distribution in each footpath environment under gender differentiation (test H3). The different influences of the factors were compared by the *p*-values and standardized correlation coefficients in the models. The indicators of the independent variables were the same in the three models (Table 1), and the structure of the multiple linear regression model is [63,64]:$$Y_{a}=\beta_{0}+\beta_{1}x_{1a}+\beta_{2}x_{2a}+\cdots +\beta_{k}x_{ka}+\varepsilon_{a}$$
where Y is the dependent variable; x denotes the independent variable; $\beta_{0}$, $\beta_{1}$, …, $\beta_{k}$ are parameters to be determined; $\varepsilon_{a}$ is a random variable and a represents the observation group value.

## 3. Results

### 3.1. Descriptive Statistics

There were differences in the frequency of JA performed by residents on different footpaths. The JA frequency of the total population for each footpath from high to low was 5483 times for community footpaths, 1506 times for urban footpaths, and 1322 times for green land footpaths (Table 2), indicating that community footpaths are the most frequently chosen for exercise by residents. Most residents are less likely to run on urban footpaths and green land footpaths, probably because the two footpaths have longer distances compared to community footpaths, and most people lack both time and energy. Among them, the frequency of JA in urban footpaths and green land footpaths is similar, and the gap between the female activity percentage and male activity percentage in these two footpaths is only 2.84% and 1.88%, respectively; the JA of residents in community footpaths is relatively larger, and the difference ratio between men’s and women’s activities is also higher, reaching 13.65%. In general, the frequency of female JA in each footpath was significantly higher than the frequency of male activity.

In terms of spatial distribution (Figure 4), the jogging trajectories of women in the road network gradually decrease outward along the center of the area, while the jogging trajectories of men are scattered in small “clusters”. The jogging lengths of urban roads with important traffic functions, such as urban trunk roads and highways, are shorter, while the jogging lengths of small roads such as community footpaths are longer. The female jogging footpath showed a continuous downward trend along the main roads of the community, extending outward from the center of the area, and still remained high on the main roads in the periphery of the area. The male jogging trajectory in the regional center has an upward-facing crescent-shaped activity trajectory, and the activity distances in most of the footpaths on the periphery of the region are very short or scattered. From the viewpoint of the gender ratio of JA, the footpath sections with high proportions of female activity are concentrated in the regional center and gradually decrease when closer to the periphery of the city, but the footpath sections with high proportions of male trips are in the regional central ring.

From the point of view of time distribution (Figure 5), the trend of jogging distance with time is more consistent for different synchronous footpaths and different genders, which is relatively consistent with the time of daily activities. From 5:00 a.m., the number of runners in each footpath network starts to rise, especially between 6:00 and 7:00, and then fluctuates slowly. The number rises abruptly between 17:00 and 20:00, and finally drops sharply at 23:00. It can be seen from Figure 4 that the two peaks of the total length of JA are during 6:00–7:00 and 18:00–20:00. During these time periods, the number of runners was highest during the evening rush hour.

### 3.2. Identification of Environmental Factors of Footpath Construction Affecting Residents’ JA

First, Pearson correlation analysis was used to identify the main environmental factors that potentially affect residents’ JA. Secondly, multiple linear regression analysis was applied to identify the core variables in the related elements, and the correlation (P) was used as the measure. When *p* < 0.05, the correlation was considered to be somewhat significant; when *p* < 0.01, the correlation was considered to be extremely significant. Through correlation analysis, a total of 11 elements of the footpath environment that potentially affect residents’ JA were identified (Table 3).

In this study, the above 11 potentially relevant elements were tested for correlation (Figure 6) and were included in a multiple linear regression model to discuss the overall impact of the footpath environment on residents’ JA levels through a full-sample analysis (Table 4). The results showed that the model shows a good simulation, and the overall *p*-value of the model was less than 0.001. Through the significance test, the mean VIF was 1.87, which did not have co-integration. The standardized coefficients (regression coefficients obtained after standardizing the data) usually expressed as β (β < 0 indicates negative correlation, β > 0 indicates positive correlation, and β closer to 1 indicates a stronger relationship) were included in the evaluation system, and it was found that SRD (β = 0.083, *p* < 0.001), BRD (β = 0.048, *p* < 0.001), and TEMP (β = 0.069, *p* < 0.001) positively promoted residents’ JA, and RLD (β = −0.03, *p* < 0.05), FD (β = −0.057, *p* < 0.01), SL (β = −0.039, *p* < 0.01), PD (β = −0.045, *p* < 0.01), BSD (β = −0.035, *p* < 0.05), and BDD (β = −0.232, *p* < 0.001) were negatively associated with residents’ JA. In this regard, hypothesis 1 of this study holds true.

### 3.3. Results of Footpath Differentiation Stage

From the perspective of trajectory differentiation, the footpath is divided into three types in this study: urban footpaths, community footpaths, and green land footpaths. The grouped multiple linear regression methods are used to analyze the effects of different footpath environment elements on the JA of residents. The $R^{2}$ of the three models were 0.790, 0.769, and 0.875, respectively, and the model had a good simulation effect. The *p*-values of the models were all less than 0.001, respectively, which all passed the test. In this regard, hypothesis 2 of this study holds.

It is found (as shown in Table 5) that SRD (β = 0.312, *p* < 0.001), BRD (β = 0.472, *p* < 0.001), and BSD (β = 0.382, *p* < 0.001) among the elements of urban footpath environment are significantly and positively related to residents’ JA in model 1; RLD (β = −0.135, *p* < 0.005), FD (β = −0.249, *p* < 0.001), TEMP (β = −0.057, *p* < 0.005), and SL (β = −0.088, *p* < 0.01) were significantly and negatively correlated with residents’ JA. In addition, although PD was poorly correlated with residents’ JA (*p* > 0.05, *p* < 0.1), it still had an effect on residents’ JA (β = 0.129) and could still be seen as a weakly correlated variable for promoting residents’ JA.

In model 2, the community footpath environment factors ARD (β = −0.15, *p* < 0.001), FD (β = −0.12, *p* < 0.001), SL (β = −0.091, *p* < 0.001), and BDD (β = −0.061, *p* < 0.01) were negatively associated with residents’ JA; RLD (β = 0.159, *p* < 0.001), GD (β = 0.29, *p* < 0.001), SRD (β = 0.08, *p* < 0.001), BRD (β = 0.597, *p* < 0.001), and BSD (β = 0.235, *p* < 0.001) were significantly and positively associated with residents’ JA. 

In model 3, among the factors of green land footpaths, RLD (β = 0.318, *p* < 0.001), GD (β = 0.122, *p* < 0.01), ARD (β = 0.49, *p* < 0.001), PD (β = 0.198, *p* < 0.001), and BDD (β = 0.155, *p* < 0.05) were significantly and positively correlated with residents’ JA. TEMP (β = −0.067, *p* < 0.01) and SL (β = −0.052, *p* < 0.05) were significantly negatively correlated with residents’ JA.

### 3.4. Footpath Differentiation and Gender Difference Stage Results

Then, we added gender to the models of the original regression analysis to verify whether they had a significant effect on different footpaths of socio-spatial differentiation (Table 6). The R^2^ of the six models were 0.910, 0.894, 0.804, 0.764, 0.852, and 0.977, respectively, and the models were simulated well. The *p*-values of the models were all less than 0.001. All of these values passed the test. In this regard, hypothesis 3 of this study holds.

It is found that RLD (β = 0.422, *p* < 0.001), SRD (β = 0.146, *p* < 0.01), BRD (β = 0.458, *p* < 0.001), BSD (β = 0.145, *p* < 0.001) and BDD (β = 0.196, *p* < 0.001) in model 1 had significant positive effects on female JA; ARD (β = −0.29, *p* < 0.001) and SL (β = −0.101, *p* < 0.001) had significant negative effects on female JA.

In model 1b, ARD (β = 0.596, *p* < 0.001), BRD (β = 0.571, *p* < 0.001), and BSD (β = 0.838, *p* < 0.001) showed significant positive effects with male JA; GD (β = −0.086, *p* < 0.05) and FD (β = −0.784, *p* < 0.01) showed significant negative effects with male JA. In addition, although the association between BDD and residents’ JA was poor (*p* > 0.05, *p* < 0.1), it still had a negative effect on residents’ JA (β = −0.118) and could be considered a weak correlation variable for inhabiting male JA.

In model 2a, RLD (β = 0.123, *p* < 0.01), GD (β = 0.15, *p* < 0.001), ARD (β = 0.184, *p* < 0.001), SRD (β = 0.062, *p* < 0.05), BRD (β = 0.478, *p* < 0.001), and BSD (β = 0.343, *p* < 0.001) associated with female JA showed significant positive effects; FD (β = −0.325, *p* < 0.001), SL (β = −0.074, *p* < 0.001), and BDD (β = −0.089, *p* < 0.001) showed significant negative effects with female JA.

RLD (β = 0.398, *p* < 0.001), GD (β = 0.484, *p* < 0.001), SRD (β = 0.079, *p* < 0.05), and BRD (β = 0.525, *p* < 0.001) in model 2b showed significant positive effects with male JA; ARD (β = −0.428, *p* < 0.001) and SL (β = −0.102, *p* < 0.001) showed a significant negative effect with male JA. In addition, despite the poor association between PD and resident JA (*p* > 0.05, *p* < 0.1), PD still positively influenced resident JA (β = 0.108) and could still be considered as a weakly correlated variable for promoting male JA.

In model 3a, GD (β = 0.208, *p* < 0.001), SRD (β = 0.378, *p* < 0.001), PD (β = 0.463, *p* < 0.001), BSD (β = 0.135, *p* < 0.05), and BDD (β = 0.238, *p* < 0.001) showed significant positive effects with female JA; only FD (β =−0.283, *p* < 0.001) showed a significant negative effect with female JA. In addition, although RLD and ARD were poorly correlated with resident JA (*p* > 0.05, *p* < 0.1), they had positive effects on resident JA (β = 0.122, β = 0.21) and could still be seen as weakly correlated variables to promote female JA; meanwhile, TEMP also had some weak correlation effects to inhibit female JA (β = −0.056).

In model 3b, RLD (β = 0.58, *p* < 0.001), ARD (β = 0.541, *p* < 0.001), FD (β = 0.377, *p* < 0.001), and BDD (β = 0.12, *p* < 0.001) showed a significant positive effect on male JA; TEMP (β = −0.033, *p* < 0.05), PD (β = −0.163, *p* < 0.01), and BSD (β = −0.43, *p* < 0.001) showed significant negative effects with male JA.

## 4. Discussion

### 4.1. Overall Stage

The importance of the environment and the PA of residents have been noticed in previous studies, and the results of the first question in this study also identified an influential relationship between the two. This study found that SRD and BRD can positively promote the JA of residents, while main roads did not show a positive and significant effect on JA. This result can be explained by the fact that main roads are often motorized spaces with many vehicles, which have a deterrent effect on residents’ outdoor jogging and easily lead to the traffic insecurity of exercisers. However, secondary roads and feeder roads are more pleasant in scale for enough exercise space and are more suitable for the PA of exercisers, which is consistent with the results of this study [55]. With the increase in FD, the study found that the JA intensity of residents decreases significantly, which is contrary to the findings of some European and American cities [56,60,65]. This may be related to the relatively compact built environment of Chinese cities and the overall higher land-use intensity in Shanghai. The concentration of high-density elements will lead to small spaces and safety issues. Therefore, the significant negative correlation between facility density and residents’ JA intensity is revealed in the results. BSD and BDD in the built environment also had a significant negative effect on residents’ JA, which has not been described before. In previous studies, accessibility is a positive factor for residents’ travel activities. The two different results may be attributed to the fact that this study aimed at physical exercise such as jogging. It is more suitable for free jogging exercise in areas with an open environment and fewer vehicles or traveling people, while previous studies focused more on residents’ travel purposes. In addition, unlike previous studies, social and natural environment factors were added to this study to comprehensively observe the factors affecting residents’ jogging exercise in multiple aspects. The results clearly show that population density as a social environment factor and ground slopes as a natural environment factor have negative effects on JA, which can be ascribed to the large population concentration and the small roads that are unsuitable for comfortable JA [66]; the large ground slope easily causes the exerciser to become tired and require more energy and physical strength to maintain JA. Meanwhile, the difficult geographical environment will demoralize the residents, so the jogging length will be reduced abruptly, and they would not be willing to perform further jogging exercise [67,68]. The most interesting finding so far is the positive effect of temperature on residents’ JA. According to the statistical analysis of the whole sample, the results show that joggers’ exercise time is mainly in spring and autumn, most of which is concentrated in spring. The average temperature in Shanghai is 5~12 °C in spring and 20.0~28.0 °C in autumn. Therefore, temperature increase has a positive effect on jogging comfort (Figure 7).

### 4.2. Footpath Differentiation Stage

Previous studies have mainly conducted questionnaire surveys or experiments to obtain the time and frequency of residents’ activities in one or part of the communities for macroscopic studies [69,70,71]. Fewer studies have been conducted to obtain activity environment elements and objective activity time and distance data from residents’ jogging trajectories. The study area is dominated by residential and employment areas with a lack of attention to the footpath network differentiation during movement and to the segregation of city footpath networks formed by different mobility levels and spatial layouts. In this regard, the second issue of this study is to investigate the influence of different footpath network environments on the spatial differentiation of JA. Therefore, the residents’ jogging environment was divided into three types of city environments: urban footpaths, community footpaths, and green land footpaths. The results showed (Figure 8) that RLD and GD both have a significant positive impact on JA only in community footpath and green land footpath environments, which further proves that community residential density and green space density are complementary and mutually beneficial [72]. Low population density will reduce the scale of travel, and high green land density represents a beautiful environment around the community and the proximity of green land to the community, which is more attractive for residents to go for JAs. 

Moreover, PD in the social environment also positively contributes to JA on green land footpaths [73,74], confirming the interaction of two environmental factors, RLD and GD. An interesting finding is that the arterial roads contribute to JA on green land footpaths, which has not been described before. This finding may be explained by the fact that green land parks border or contain arterials. Such green land parks are mainly large-scale activity sites such as city center parks or commercial parks with dense arterial roads, which can drive more tourists or residents to travel to this activity. However, the conclusion is slightly different from our usual perception and needs to be further verified. 

Surprisingly, TEMP had a negative effect on JA in urban footpaths and green land footpaths, which is different from the full sample study. According to the statistical analysis of urban footpath samples and green land footpath samples (Figure 9), the results show that the sports time of the two is mainly in summer and autumn, more than in spring and winter. The average temperature in Shanghai is 5~12 °C in spring, 21.0~28.0 °C in summer, 20.0~28.0 °C in autumn, and 4~12 °C in winter. Therefore, the temperature increase has a negative effect on jogging comfort. However, due to the small sample size, the research conclusion needs further research and verification. 

The study also confirmed that BSD had a positive impact on JA for urban footpaths and community footpaths, and BDD had a negative impact on JA for community footpaths, which is consistent with the findings of most scholars that the accessibility is beneficial to the JA of residents [75,76]. The greater the number of bus stops, the closer the distance from the bus stop, indicating that transportation is more convenient and more attractive for residents to go out [77]. In addition, BDD had a positive impact on the JA of the green land footpaths. The farther away from the bus stop, the higher the intensity of residents’ activities because the closer the bus stop, the more noise and exhaust emissions generated by the passing of buses will cause serious damage to the green land [78,79].

### 4.3. Footpath Differentiation and Gender Difference Stage

Based on the analysis of the above two models, six multiple linear regression models were constructed separately to further explore the influence of footpath differentiation on JA under gender differences. As shown in Figure 10, there are also significant differences in the effects of different footpath environments on the JA of different gender groups. An interesting finding was that green land density had a significant negative effect on male JA on urban footpaths, whereas main roads had a positive effect on male JA on green land footpaths. The two results of this study come from two different statistical ranges and analytical models: the former statistical range is a buffer zone extended by 500 m centered on the movement trajectory of joggers on urban footpaths; the latter statistical range is a buffer zone expanded by 500 m centered on the movement trajectory of joggers on green land footpaths. Therefore, we think these two results should be explained separately. The former result shows that the green land space density within 500 m on both sides of the urban footpath has a weak negative correlation with the JA of males. It may be that the green land belt concentration area is generally an urban park, which is less open to the urban footpath directly, resulting in low accessibility from the urban footpath to the urban park; however, much of the low-density green land on both sides of the urban footpaths is ancillary green land of urban roads, which is directly open to the urban footpath, leading to higher accessibility than the urban park, and thus they are more attractive to male joggers. The latter result shows that the density of urban main roads within 500 m on both sides of the green land footpaths has a strong positive correlation with the JA of males. It may be that green land includes all kinds of green land footpaths, and the density of nearby main roads can directly affect the accessibility of joggers. 

However, ARD induces opposite results for different gender groups jogging on urban footpaths; ARD has a negative effect on women jogging on urban footpaths but a positive effect on women jogging on community footpaths. However, ARD has a completely different effect on men. The above finding may be attributed to the fact that the denser main roads carry more vehicles, which poses a certain threat to the safety of women exercising on urban trails, and thus there is a certain negative effect on women jogging outdoors [48]; the environmental factors that have a positive effect on women jogging on community footpaths are similar to some scholars’ studies. Their study showed that there are usually green landscaped areas between community and arterial roads, and landscaped areas provide spaces for residents to rest. The shadows under the trees in landscaped areas are precisely projected on the resting space, creating a good resting place for the residents. Female groups usually prefer green land with beautiful landscapes and rich activities, which may be one of the reasons why green lands promote jogging in female groups [80]. In contrast, the male group likes to run in quiet spaces, and the landscape of green areas can play a certain attenuating effect on the traffic noise generated by arterial roads. However, excessive traffic noise pollution from dense arterial roads still cannot be completely eliminated, and thus dense arterial roads will have a negative impact on men who run on community footpaths [81]. 

An unexpected finding is that the number of public service facilities has an opposite influence on the JA of different gender groups on green land footpaths, inhibiting the JA of female groups but promoting the JA of male groups there. The average single jogging distance and duration of men and women on the green land footpaths were counted: the females’ average jogging duration is 1.10 h, whereas the males’ average jogging duration is 1.60 h. The females’ average jogging distance length is 7.79 km, and the males’ average jogging distance length is 11.30 km. These comparison results show that compared with males, females’ average single jogging durations and distances are shorter, and they may have more chances to stop and contact surrounding service facilities. In addition, the social environmental factor, PD, and the natural environmental factors, TEMP and SL, were also shown to have significant effects on the JA of the different gender groups in the six-category model. Therefore, the influence of these elements on them can be considered in future environmental planning.

## 5. Conclusions

This study aims to analyze the influence of footpath environments on the JA of residents. Grouped multiple linear regression models were used to demonstrate the differences in the influence of different footpath environments on the JA of residents and different gender groups. The conclusions obtained are as follows:

(1) Jogging activities are concentrated in the community footpath environment and reach their peak at night; (2) The rise and fall of environment elements (the built environment, social environment, and natural environment) significantly affects the relative jogging length of residents; (3) RLD has a positive impact on the JA in both community and green land footpaths and has a negative impact on the JA of urban footpaths. However, ARD and BDD had opposite significant effects on the JA on both community and green land footpaths; (4) ARD had opposite and significant effects on the JA of residents of different genders on both urban and community footpaths. FD, PD, and BSD also had opposite and significant effects on the JA of residents of different genders on green land footpaths.

In summary, the results of this study found that the same planning criteria cannot be adopted for different types of footpaths, and we propose a theory for a method of identifying footpath environments. The specific application of this theory is as follows: Firstly, different types of city footpaths are laid out with varying configurations of environmental elements to promote JA. In the community and green land footpath environments, as much as possible to increase the density of residential and green land to encourage JA, but in the urban footpath environment, the environment should avoid the negative impact of excessive residential construction on JA. ARD and BSD should be increased in the green land footpaths’ surrounding environments to promote JA. Nonetheless, there is a need to reduce the ARD and BSD in the community footpath environment to avoid adverse effects on JA. The study also showed that men and women have different needs for footpaths. Females preferred ARD environmental elements in community footpaths and PD and BSD environmental elements in green land footpaths. At the same time, males had higher requirements for ARD environmental elements in urban footpaths and FD environmental elements in green land footpaths. In addition, the sports data in this paper show that men are less motivated to go jogging than women; to enhance and promote the attractiveness of outdoor jogging for different gender groups, the layout of varying footpath environment configurations should be carried out according to the differential demands of environmental elements for residents of different genders to promote social and gender equity in exercise health.

In this paper, there are some shortcomings in the research process. This study only uses the data of JA, which is a limitation in the data analysis. In addition, the discussion about the influence mechanism of footpath differentiation on residents’ jogging is mainly based on the model and analysis of the existing related literature and the authors’ experiences, lacking a large number of qualitative investigations. In addition, this study mainly uses natural experimental data, and the lack of environmental perception data, population, and economic census data is also one of the shortcomings of this study. Meanwhile, the data of our research are from urban areas, and there is a lack of access to data in other rural and remote areas in this study. We hope that the follow-up will further deepen the exploration of these aspects.

## Figures and Tables

**Figure 1 ijerph-19-14372-f001:**
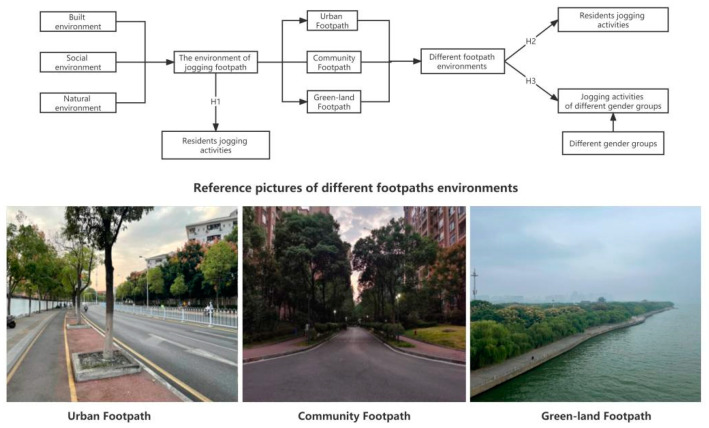
Theoretical Hypothesis Framework.

**Figure 2 ijerph-19-14372-f002:**
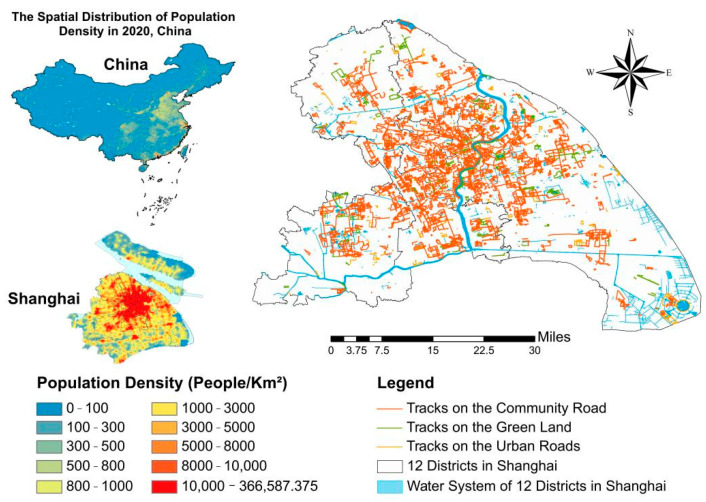
Map of the study area.

**Figure 3 ijerph-19-14372-f003:**
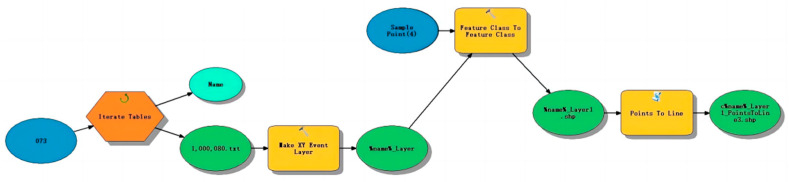
Motion trajectories in ArcGIS software.

**Figure 4 ijerph-19-14372-f004:**
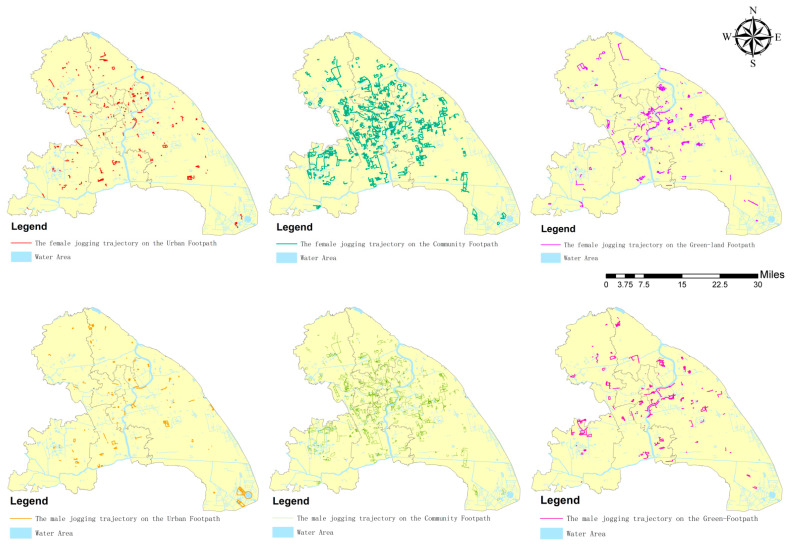
Distribution of JA on each footpath for different gender.

**Figure 5 ijerph-19-14372-f005:**
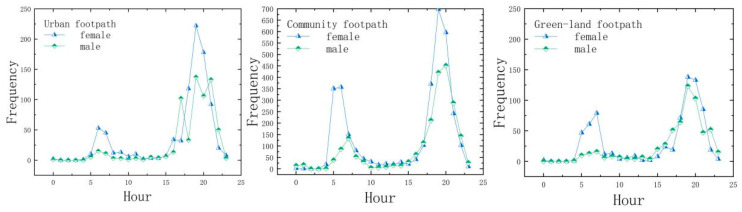
Twenty-four hours of jogging of different gender groups on each footpath.

**Figure 6 ijerph-19-14372-f006:**
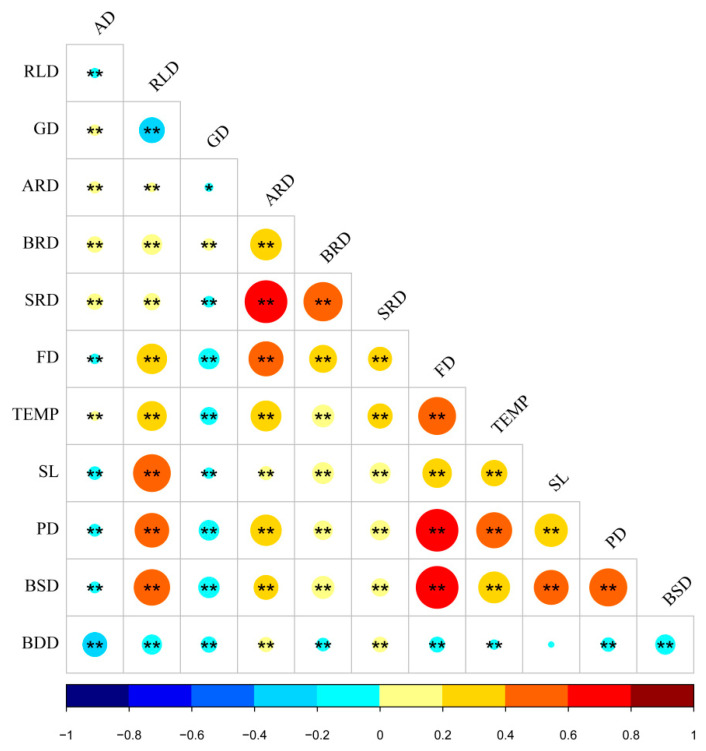
Correlation test of footpath environment elements. Note: “−1 to 1” represents the range of correlation path coefficient values; The size of the circle represents the size of the path coefficient value; “**” indicates *p* < 0.01, “*” indicates *p* < 0.05.

**Figure 7 ijerph-19-14372-f007:**
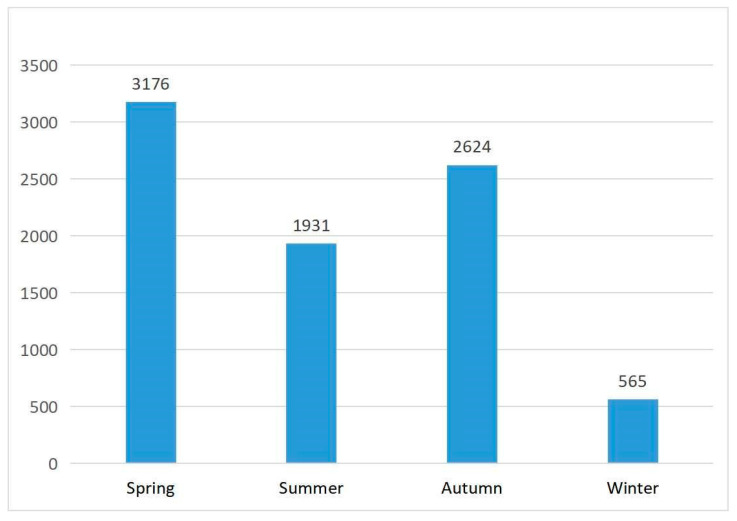
Statistics of the number of jogging trajectories in different seasons (full-sampled).

**Figure 8 ijerph-19-14372-f008:**
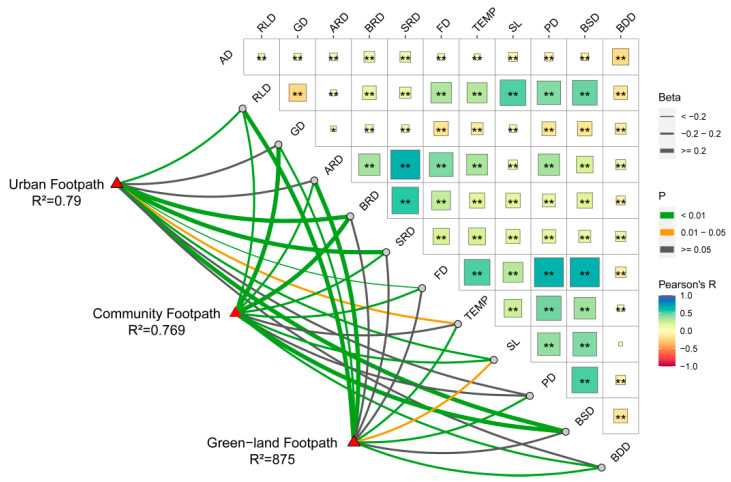
Differences in the influence of different footpath environments on residents’ JA. Note: “**” indicates *p* < 0.01, “*” indicates *p* < 0.05.

**Figure 9 ijerph-19-14372-f009:**
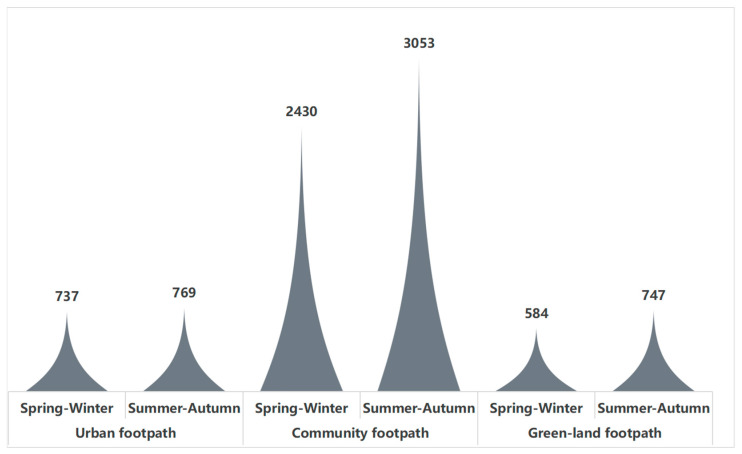
Statistics of the number of jogging trajectories in different seasons (different types of footpaths).

**Figure 10 ijerph-19-14372-f010:**
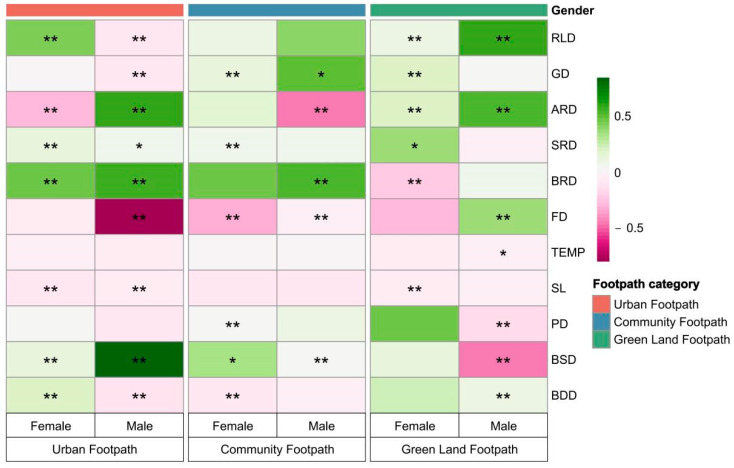
Heat map of the significance comparison of different footpath environments on the JA of different gender groups. Note: green represents a positive value; purple represents a negative value; the depth of the color represents the size of the normalization coefficient; “**” indicates *p* < 0.01, “*” indicates *p* < 0.05.

**Table 1 ijerph-19-14372-t001:** Footpath environmental indicator system.

Category	Variable	Calculation Method	Abbreviation
Independent	Residential land density	Residential land area/Buffer zone area	RLD
	Green land density	Green land area/Buffer zone area	GD
	Nighttime light	Sum of average units/Buffer zone area	NTL
	Arterial road density	Arterial road quantity/Buffer zone area	ARD
	Secondary road density	Secondary road quantity/Buffer zone area	SRD
	Branch road density	Branch road quantity/Buffer zone area	BRD
	Land-use mix	Land-use mix area/Buffer zone area	LM
	Facilities diversity	The number of POIS in 13 types of facilities, including restaurants, scenic spots services, public services, enterprises, shopping services, transportation, financial services, education and culture services, commercial housings, life services, sports and leisure services, medical treatment, and government services/Buffer zone area	FD
	Metro line density	The length of the subway line/Buffer zone area	MLD
	Bus stop density	Bus stop quantity/Buffer zone area	BSD
	Precipitation	Sum of average units/Buffer zone area	PR
	Temperature	Sum of average units/Buffer zone area	TEMP
	Normalized difference vegetation index	Sum of average units/Buffer zone area	NDVI
	Slope	Sum of average units/Buffer zone area	SL
	Bus distance density	Distance to the bus stop/Buffer zone area	BDD
	Metro distance density	Distance to the metro stop/Buffer zone area	MDD
	Population density	Sum of average units/Buffer zone area	PD
	Gross Domestic Product	Sum of average units/Buffer zone area	GDP
Dependent	Activity distance	The total jogging distance of a single person in the buffer zone	AD

**Table 2 ijerph-19-14372-t002:** Distribution of residents’ JA trajectory.

Type		Female Activity	Male Activity	Female-Male Ratio
Total Value	frequency	4919	3392	
	percentage	59.20%	40.80%	145.02%
Urban footpath	frequency	871	635	
	percentage	10.48%	7.64%	137.17%
Community Footpath	frequency	3309	2174	
	percentage	39.81%	26.16%	152.21%
Green land Footpath	frequency	739	583	
	percentage	8.89%	7.01%	126.76%

**Table 3 ijerph-19-14372-t003:** Correlation analysis between footpath environment and residents’ JA.

Variable	β	*p*	Influence
RLD	−0.034	0.002	**
NTL	0.014	0.2	—
GD	0.041	<0.001	**
ARD	0.048	<0.001	**
SRD	0.089	<0.001	**
BRD	0.090	<0.001	**
FD	−0.036	0.001	**
LM	0.015	0.163	—
PR	0.002	0.821	—
TEMP	0.030	0.007	**
NDVI	0.002	0.843	—
SL	−0.063	<0.001	**
PD	−0.055	<0.001	**
GDP	−0.014	0.198	—
MLD	0.017	0.125	—
BSD	−0.046	<0.001	**
MDD	−0.021	0.059	—
BDD	−0.211	<0.001	**

Note: “**” indicates *p* < 0.01, and “—” indicates that the correlation is not significant. RLD, Residential land density; NTL, Nighttime light; GD, Green land density; ARD, Arterial road density; SRD, Secondary road density; BRD, Branch road density; FD, Facilities diversity; LM, Land-use mix; PR, Precipitation; TEMP, Temperature; NDVI, Normalized difference vegetation index; SL, Slope; PD, Population density; GDP, Gross Domestic Product; MLD, Metro line density; BSD, Bus stop density; MDD, Metro distance density; BDD, Bus distance density.

**Table 4 ijerph-19-14372-t004:** Multiple linear regression analysis and multicollinearity test of footpath environment and residents’ JA.

Variable	β	*p*	VIF	Tolerance Value
RLD	−0.03	0.034 *	1.743	0.574
GD	−0.001	0.955	1.125	0.889
ARD	0.023	0.161	2.417	0.414
SRD	0.083	<0.001 ***	2.482	0.403
BRD	0.048	<0.001 ***	1.576	0.635
FD	−0.057	0.001 **	2.882	0.347
TEMP	0.069	<0.001 ***	1.479	0.676
SL	−0.039	0.003 **	1.567	0.638
PD	−0.045	0.003 **	2.115	0.473
BSD	−0.035	0.023 *	2.152	0.465
BDD	−0.232	<0.001 ***	1.085	0.922
*p*	<0.001 ***			
*R* ^2^	0.725			

Note: “*”, “**”, “***” pass the test at 0.05, 0.01, and 0.001 significance levels, respectively. RLD, Residential land density; GD, Green land density; ARD, Arterial road density; SRD, Secondary road density; BRD, Branch road density; FD, Facilities diversity; TEMP, Temperature; SL, Slope; PD, Population density; BSD, Bus stop density; BDD, Bus distance density.

**Table 5 ijerph-19-14372-t005:** Effects of different footpath environments on residents’ JA.

Variable	Model 1: Urban Footpath		Model 2: Community Footpath		Model 3: Green Land Footpath	
	β	*p*	β	*p*	β	*p*
RLD	−0.135	0.014 *	0.159	<0.001 ***	0.318	<0.001 ***
GD	−0.03	0.253	0.29	<0.001 ***	0.122	0.002 **
ARD	0.086	0.188	−0.15	<0.001 ***	0.49	<0.001 ***
SRD	0.312	<0.001 ***	0.08	<0.001 ***	0.017	0.802
BRD	0.472	<0.001 ***	0.597	<0.001 ***	−0.105	0.25
FD	−0.249	<0.001 ***	−0.12	<0.001 ***	0.033	0.646
TEMP	−0.057	0.037 *	0.003	0.845	−0.067	0.003 **
SL	−0.088	0.001 **	−0.091	<0.001 ***	−0.052	0.018 *
PD	0.129	0.059	0.034	0.249	0.198	<0.001 ***
BSD	0.382	<0.001 ***	0.235	<0.001 ***	−0.067	0.22
BDD	0.036	0.271	−0.061	0.001 **	0.155	<0.001 ***
*p*	<0.001 ***		<0.001 ***		<0.001 ***	
*R* ^2^	0.79		0.769		0.875	

Note: “*”, “**”, “***” pass the test at 0.05, 0.01, and 0.001 significance levels, respectively. RLD, Residential land density; GD, Green land density; ARD, Arterial road density; SRD, Secondary road density; BRD, Branch road density; FD, Facilities diversity; TEMP, Temperature; SL, Slope; PD, Population density; BSD, Bus stop density; BDD, Bus distance density.

**Table 6 ijerph-19-14372-t006:** Effect of different footpath environments on JA in different gender groups.

Variable	Model 1: Urban Footpath		Model 2: Community Footpath		Model 3: Green Land Footpath	
	Female	Male	Female	Male	Female	Male
RLD	0.422 ***	−0.08	0.123 **	0.398 ***	0.122	0.58 ***
GD	0.013	−0.086 *	0.15 ***	0.484 ***	0.208 ***	0.038
ARD	−0.29 ***	0.596 ***	0.184 ***	−0.428 ***	0.21	0.541 ***
SRD	0.146 **	0.07	0.062 *	0.079 *	0.378 ***	−0.023
BRD	0.458 ***	0.571 ***	0.478 ***	0.525 ***	−0.23	0.065
FD	−0.039	−0.784 **	−0.325 ***	−0.015	−0.283 ***	0.377 ***
TEMP	−0.023	−0.052	0.004	0.008	−0.056	−0.033 *
SL	−0.101 ***	−0.046	−0.074 ***	−0.102 ***	−0.042	−0.017
PD	0.03	−0.099	0.05	0.108	0.463 ***	−0.163 **
BSD	0.145 ***	0.838 ***	0.343 ***	0.041	0.135 *	−0.43 ***
BDD	0.196 ***	−0.118	−0.089 ***	−0.011	0.238 ***	0.12 ***
*p*	<0.001 ***	−0.08	<0.001 ***	<0.001 ***	<0.001 ***	<0.001 ***
*R* ^2^	0.910	0.894	0.804	0.764	0.852	0.977

Note: “*”, “**”, “***” pass the test at the significance level of 0.05, 0.01, and 0.001, respectively. RLD, Residential land density; GD, Green land density; ARD, Arterial road density; SRD, Secondary road density; BRD, Branch road density; FD, Facilities diversity; TEMP, Temperature; SL, Slope; PD, Population density; BSD, Bus stop density; BDD, Bus distance density.

## Data Availability

The data that support the findings of this study are available from the Dorray Sports app. Restrictions apply to the availability of these data, which were used under license for this study.

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
