# Peer review of "How Do Different Urban Footpath Environments Affect the Jogging Preferences of Residents of Different Genders? Empirical Research Based on Trajectory Data"

_ijerph, 2022, doi:10.3390/ijerph192114372_

Round 1

Reviewer 1 Report (Previous Reviewer 3)

Dear Authors

Thank you for revising and resubmitting the revised version, which has greatly improved readability, presentation quality, and writing logic. Therefore, I recommend the publication of this valuable study.

However, please check the manuscript again before publication, as Figure 4 does not seem to be shown correctly.

Also, please forgive me for my concern about some trivial issues. The authors' response to comment 5 (the impact of the number of public service facilities on JA for different gender groups) is unsatisfactory because the authors chose to remove statements rather than rethink the reasons (which seems to me to be a cop-out rather than a response to my comment). For me personally, it is the discussion and analysis in response to some of the counterintuitive results that is the most valuable aspect of social science research.

Overall, of course, I applaud the authors for their hard work and efforts.

Good luck and congratulations.

Author Response

Dear reviewer:

Thanks very much for taking the time to review this manuscript. We appreciate all your generous comments and suggestions! Please find my itemized responses in below and my revisions in the re-submitted files.

Sincerely yours,

Qikang Zhong, Bo Li, Yue Chen.

Reviewer 2 Report (New Reviewer)

Overall, this study of the gender differences relative to footpath types is sound. The authors do an excellent job of managing environmental variables like temperature, building density, etc. Still, they do not account adequately for possible socio-psychological differences that may account for preferred path selection based on gender. For example, women may choose a path based on the number of other people on the path related to perceptions of personal safety. The cognitive factors generally associated with safety are legibility, enclosure, complexity, crime potential, wildlife, and lighting. The research did not include these factors and it needs to be clear that it excludes cognitive measures and focuses solely on environmental conditions. 

Author Response

Dear reviewer:

Thanks very much for taking the time to review this manuscript. We appreciate all your generous comments and suggestions! Please find my itemized responses in below and my revisions in the re-submitted files.

Sincerely yours,

Qikang Zhong, Bo Li, Yue Chen.

Reviewer 3 Report (New Reviewer)

The manuscript is well written and organized. The argument is original and aligned with the scope of the journal. According to my opinion it could be accepted for publication after minor improvements.

Tables 7-9 are too concise. Maybe the data contained could be reported differently.

Figures like 7 are beautiful. However, a bit complicated to understand. I would enhance them by introducing them in a more extensive and more explicit way.

The choice of the statistical model used could be better justified.

Could a model like this be extended to analyse vehicle and logistics routes? In my opinion, yes. I would therefore also enhance the manuscript in this direction, referring to how this could improve eco-design and sustainability, also in teaching. For example, refer to: Spreafico, C., & Landi, D. (2022). Investigating students’ eco-misperceptions in applying eco-design methods. Journal of Cleaner Production, 342, 130866.

Author Response

Dear reviewer:

Thanks very much for taking the time to review this manuscript. We appreciate all your generous comments and suggestions! Please find my itemized responses in below and my revisions in the re-submitted files.

Sincerely yours,

Qikang Zhong, Bo Li, Yue Chen.

This manuscript is a resubmission of an earlier submission. The following is a list of the peer review reports and author responses from that submission.

Round 1

Reviewer 1 Report

I would like to thank the authors for the opportunity to review this manuscript. It covers a novel topic and uses an innovative method to investigate an under-researched area. However, there are several grammatical errors throughout the paper and substantial editing of the English is needed. Additionally, in many parts of the manuscript, key concepts and points could be better organised for a paper that is more readable and easy to follow. Further editing is needed to improve this manuscript to a state where it could be published. Some specific comments are provided below for improving this paper. 

ABSTRACT:

In the abstract, the use of several abbreviations is hard to follow and there are some grammatical errors throughout.

INTRODUCTION:

-Page 1, line 29: what is meant by "simple PA"? Are you referring to "simply engaging in PA"?

-Page 1, lines 32: could the authors speak more about the literature showing the impact of the urban environment on health outcomes?

-Page 1, lines 35-37: Can you expand on how jogging is one of the most beneficial daily exercise modes? For example, most beneficial in reducing chronic diseases? More detail is needed.

-Page 2, lines 62-64: further explanation of the relevant theory is needed

-Page 2 lines 62-62: is there any existing evidence to suggest this?

-Page 2 lines 5-58: could the authors expand upon the potential influence of social and natural environmental factors on jogging activities? This is a bit vague

Introduction: There are aspects of the Intro that are a bit vague and that needs to be more specific and further supported by existing literature. Additionally, parts of the Introduction are a big segmented and hard to follow and better organisation is needed for improved clarity about the background of this topic. Additionally, there are many grammatical errors and the Introduction would benefit from substantial editing in English and wording.

-Methods: Page 6 (2.3 Analysis method), line 180: The exact SPSS version used should be stated

-Methods: Page 6 (2.3 Analysis method): the authors should specify whether there models were unadjusted or adjusted.

-Page 6 lines 204-206: This statement could be corroborated if supported by any relevant literature

-Methods: This study does a nice job a employing novel methods to explore a gap in the literature and an innovative research topic. At times, further detail is needed, particularly when describing the data analyses.

-Results: Page 6: further information about the sample is needed, such as age of participants.

-Results: The heavy use of abbreviations throughout the results makes it a bit harder to follow the results and the findings. Under the tables (e.g., Tables 3, 4, 5, 6), the full variable names should be listed for the readers. Alternatively, the abbreviated variable names can remain in the tables as is, but the written part of the results should use full variable names.

-page 16 lines 532-538: The authors should also discuss the validity of using the specific wearable device to assess physical activity and provide commentary around it compares to other measure of physical activity/jogging, such as accelerometers. It should also be noted that the findings may not be generalisable to other rural and remote areas in China as well as other countries (including developed and developing). The limitations of the study design should also be noted. 

-Results: Very long paragraphs are used in each section of the results, which makes this part harder to read. Further organisation and dividing of these sections would enhance the readability.

-Conclusions: the summary of the findings could be enhanced by using more succinct writing

Author Response

Dear reviewer:

Thanks very much for taking your time to review this manuscript. We appreciate all your generous comments and suggestions! Please find my itemized responses in below and my revisions in the re-submitted files.

Reviewer 2 Report

Sept. 8/22

Review  IJERPH

How do different urban footpath environments affect the jogging preferences of residents of different genders?- an empirical research based on trajectory data

I would like to commend the authors for the amount of work they did to complete this study. I believe examining physical activities using novel approaches (such as trajectory data) is highly warranted. However, I found it very difficult to comprehend several concepts expressed in the manuscript mainly due to the poor use of the English language. I also believe that beyond the problem with language usage, the authors are not writing in a clear manner – e.g., sentences and concepts do not “flow” as they should throughout the manuscript. There are also instances when important statements are supported by questionable previous studies (such as the rationale for studying running – jogging has been proved to be one of the most beneficial daily exercise modes for public health references 6, 7 from 2003 and 2007 on training effects in adults and college student health).

Author Response

Dear reviewer:

Thanks very much for taking your time to review this manuscript. We appreciate all your generous comments and suggestions! Please find my itemized responses in below and my revisions in the re-submitted files.

Kind regards,

Reviewer 3 Report

Dear Authors.

I have completed my review of this manuscript and I found it to be a valuable study with good graphical presentation and structure. However, the following issues still need to be addressed before further consideration.

Point 1 (Abstract) Usually in the abstract need to write the full name, such as RLD, BDD these parameters the reader can not understand the specific meaning in the abstract.

Point 2 (Introduction) "Whether it is fast or slow, simple PA is..."

Even if there is a full name in the abstract, the full name needs to be spelled out in the text when it first appears.

Point 3 (Introduction) "secondly, the different path environments are related to the distribution of residents' jogging trajectory, so the different path surroundings will have a differential impact on JA (H2)"

Based on the results in Table 5, the description here should be "Different path types of environments have different effects on JA."

Point 4 (Figure 1) The hypothesis is presented abruptly, what is the definition of the three footpaths? Second, I can understand the greenland path, but how do the authors distinguish the difference between urban paths and community paths? Third, the description of the hypothesis is not consistent with Figure 1, for example, the difference of gender is not involved in H1. It is suggested that the authors strictly follow the hypothesis to make Figure 1; in addition, it is suggested to provide some reference pictures to illustrate the three types of footpaths.

Point 5 (Line 124-126) "The study data obtained from a smartwatch provider (Dory Wearables), reflect the daily activities of residents (Figure 2)."

I'm very sorry, I didn't find any information about this device in the web, can the author provide information about the company of this device? Also, can the author clarify how the data was obtained? As the author states, it contains private information such as gender, so did it pass the ethical and moral review of the relevant agencies? Also, how did the authors filter the data? If the device recorded the daily activities of residents, how did the authors determine if the sample could be identified as jogging activity?

Point 6 (Line 138-139) "The processed dataset contained 8311 valid trajectory data with a sex ratio of 2.11. "

Please add the time period of the dataset, in what period were these data captured (time?date?)?

Point 7 (Line 150-151) "Therefore, in this study, 17 indicators were selected as built environment factors for jogging footpath"

These metrics were proposed abruptly and the authors did not elaborate on the theoretical background of the selection of these metrics in the Introduction section, like why do the authors believe that nighttime light density and precipitation density affect JA? Is there any literature that indicates these potential associations? Besides, only a brief description of the subjects for which data were acquired is given here. The detailed process/method of data collection (e.g., NDVI) and time (e.g., precipitation, temperature) needs to be described so that other researchers can replicate the study.

Point 8 (Line 208-209) "and the male-female ration in this two footpaths is only 2.84% and 1.88%, respectively..."

Not 137.71% and 126.76%?

Point 9 (Table 2) I suggest that the last column should be titled "Female-male ratio"

Point 10 (Figure 3) I suggest that the authors present the three paths separately; it is difficult for the reader to distinguish the three paths in the current version.

Point 11 (Figure 4) Why A-Female and A-male? what does A stand for?

Point 12 (Table 3) Note that the p-value can never be equal to 0. If the p-value is small, please indicate it as p < 0.001.

Point 13 (Line 263 and 265) Should "residential JA" be "residents' JA"?

Point 14 (Table 4) I think VIF should be a multicollinearity test rather than a covariance test; secondly, usually, there are two parameters for the multicollinearity test of multiple linear regression analysis, VIF as well as the tolerance value, please add the result of the tolerance value; furthermore, the p-value should never be equal to 0, and if the p-value is small, please indicate it as p < 0.001.

Point 15 (Line 278-279) "The p-values of the equations were 0.000, 0.003, 0.000 and 0.000..."

Why are there 4 p-values? Also, I would like to emphasize again that p-values cannot be equal to 0. Please check thoroughly in the whole text. Besides, it should be a "model" and not an "equation".

Point 16 (Line 290-292) "In addition, although PD was poorly correlated with resident JA (p > 0.05, p < 0.1), it still had an effect on resident JA (β = 0.129) and could still be seen as a weakly correlated variable for promoting resident JA"

Note that this is the result in model 1 and not in model 2.

Point 17 (Table 6) a and b should be displayed as corner labels.

Point 18 (Line 355-358) "Therefore, the significant negative correlation between residential density and residents' JA intensity are revealed in the results"

Please note that the manuscript here is talking about facility density and not residential density.

Point 19 (Line 425-427) “The proximity of green space areas to urban footpath attracts male residents to green-land footpath for PA, while men who stay on urban trails are significantly become less. ”

The authors' interpretation of some results is unconvincing. For example, in this statement, if residents enter the greenland footpath for PA because they are attracted by the green space, how do the authors explain the positive effect of GD in Model 2 of Table 5?

Point 20 (Line 457-463) "If there are too many of these public service facilities, it will drive women to maintain the family and raise children, and take on the double responsibility of wife and mother. Women will more likely to go to food markets or shopping centers to buy necessities, or to spend time with their children, rather than to green-lands for exercise. The density of service facilities, on the other hand, will bring convenience to the male group, making it easier to buy food and drinks or perform other services after their exercise."

This study wishes to discuss jogging activities rather than the daily behavior of residents, so I strongly disagree with the discussion here about the negative effects of FD on women. Would women reduce JA simply because they need to do daily chores? On the other hand, if service facilities make it easier for men to purchase food and beverages after JA, why women weren't encouraged to use them more often as a result?

Point 21 (Figure 7) The category on the right is wrong, it should be "path category" instead of "gender". Also, what does the color heat map represent? Please label it.

Point 22 (Line 492-493) "however, the appropriate increase of TEMP in the natural environment can promote residents' JA."

Based on the results in Tables 5 and 6, it seems that the effect of temperature is negative for all Green-land Footpaths.

Author Response

(The authors gave the same response as above.)

Round 2

Reviewer 3 Report

Dear Authors.

I would like to thank you for your revisions to the manuscript and your responses to the comments, which have greatly improved the quality of the manuscript. However, there are still some minor concerns that need attention.

Point 1 (Response to comment 5)  I would like to thank the author for the response and I suggest that these explanations (including figure 1-2) be added in the corresponding places in the manuscript. Also, please add a statement of data confidentiality to the Data Availability Statement at the end of the manuscript.

Point 2 (Response to comment 7 I have carefully checked the references provided by the authors (Refs. 60-62), however, there is no mention of the relationship between the selected 17 indicators and JA in these three papers. The authors are asked to consider again the question I raised in my previous comment, i.e., the theoretical basis for the selection of each indicator.

Point 3 (Response to comment 10 Thanks for the revision of Figure 4. However, I would like to express that the current version of Figure 3 (which follows Table 2) mixes all three paths, which makes the picture appear in a very confusing way and the reader cannot clearly see the distribution of the three paths. Therefore, my suggestion is that the authors should show the three paths for different genders separately in each of the six maps.

Point 4 (Response to comment 11) Thanks to the author for the revision, but I don't think it's necessary to mention it twice, please consider keeping the label in the top left corner or the figure note.

Point 5 (Response to comment 20) I am sorry, but it seems that the author did not answer my first question. As authors state that "If there are too many of these public service facilities, it will drive women to maintain the family and raise children, and take on the double responsibility of wife and mother. Women will more likely to go to food markets or shopping centers to buy necessities, or to spend time with their children, rather than to green-lands for exercise." 

This study wishes to discuss jogging activities rather than the daily behavior of residents. So, my question is "Would women reduce JA simply because they need to do daily chores?" Or, "Will women reduce their JA because of too many public service facilities?

Besides, I suggest add the table 1-1 in the corresponding places in the manuscript.

Point 6 (Response to comment 22) Thanks to the author for the response. I suggest adding these statements and the table 1-2 to the manuscript.

Author Response

Dear reviewer:

Thanks very much for taking your time to review this manuscript. We appreciate all your generous comments and suggestions! Please find my itemized responses in below and my revisions in the re-submitted files.

Kind regards,

Qikang Zhong
